# Predictors of Depression Level among Community-Dwelling Elderly Persons

**DOI:** 10.3390/ijerph19159414

**Published:** 2022-08-01

**Authors:** Chin-Chen Liu, Yi-Tung Lin, Kung-Chuan Cheng, Hsueh-Hsing Pan, Chou-Ping Chiou

**Affiliations:** 1Department of Family Medicine, E-Da Dachang Hospital, Kaohsiung City 82445, Taiwan; ed106112@edah.org.tw; 2Department of Nursing, E-Da Hospital, Kaohsiung City 82445, Taiwan; ed108027@edah.org.tw; 3Division of Colorectal Surgery, Department of Surgery, Kaohsiung Chang Gung Memorial Hospital, Chang Gung University College of Medicine, Kaohsiung City 83340, Taiwan; topguncheng@cgmh.org.tw; 4School of Nursing, National Defense Medical Center, Taipei City 11420, Taiwan; 5School of Nursing, I-Shou University, No. 8, E-Da Road, Jiau-Shu Tsuen, Yan-Chau Shiang, Kaohsiung City 84020, Taiwan

**Keywords:** elderly, social support, loneliness, depression

## Abstract

Elderly people in the community have difficulty taking care of themselves because of their inability to care for themselves in daily life as well as their poor social support system, which leads to loneliness, resulting in depression. The primary objective was to investigate the level of depression and related factors among community-dwelling elderly persons (CDEP). This was a cross-sectional study, and 150 participants were interviewed. The questionnaires included demographics, the Functional Independence Measure, the Interpersonal Support Evaluation List, the UCLA Loneliness Scale, and the Geriatric Depression Scale Short Form. It was found that participants with different levels of depression accounted for 26%, and education level, living status, chronic disease, daily life function, social support, and loneliness were all significant factors influencing the depression level among the CDEP that could significantly predict 63.4% of the variation in depression level. Nursing staff must understand the level of depression and its influencing factors, encourage the elderly in the community to increase social networks, and integrate leisure into their lives, thereby enhancing the sense of value and meaning of life and reducing feelings of loneliness and depression.

## 1. Introduction

The population of the elderly has increased substantially in most countries worldwide. The United Nations’ World Health Organization (WHO) defines people over 65 years of age as “elderly”, and when 7% of a country’s total population is over 65, it is considered an aging society. In March 2018, Taiwan’s population over 65 reached 14%, making it an “aged society”, and by the end of 2021, the population over 65 increased to 16.4%, and the aging index was 131.4% [1]. The increased proportion of the elderly has led to an increased impact of changes in bodily structure or function, along with a continuous process of irreversible deterioration [2]. The gradual decline in the physical function of the elderly leads to various diseases and disabilities, resulting in diminished activity, limited income, chronic diseases, or living alone, all of which are likely to cause psychological problems such as depression [3]. 

The WHO ranks depression, along with cardiovascular diseases and malignant tumors, as one of the three major diseases of the 21st century, owing to the severe disability it causes. Domestic research has found that depression affects about 6–38% of the elderly in Taiwan [4,5], while foreign studies have reported about 2–12% [6,7]. Depression in the elderly increases the decline in physical, psychological, social, and cognitive functions, as well as adds to caregiver burden and makes the treatment of underlying medical conditions more difficult [8]. Since depression symptoms in elderly patients are often confused with the discomfort caused by physical illnesses, depression in the elderly is often underdiagnosed [9]. When depression in community-dwelling elderly persons (CDEP) is not dealt with, it can lead to loss of meaning in life, loss of self-worth, suicide, or even death [10,11].

Loneliness refers to psychological pain caused by a lack of interpersonal intimacy. This is a serious problem for the aging population, which stems from poor health, poor material conditions, and low social support [12]. Globally, as many as 50% of the elderly are at risk of loneliness, and one-third will experience loneliness in their later years [13]. In the elderly, it can be easily overlooked, leading to increased physical and psychological problems, reduced social interactions, and impacted senses of security, intimacy, self-worth, and belonging [14,15]. Studies have also pointed out that loneliness can have a negative impact on personal health, causing depression and other mental and physical conditions [15]. 

Depression is a serious concern for the elderly. For the CDEP, higher gastrointestinal symptoms [16], reduced daily life function, and diminished social support networks lead to greater depression, which is the most common mental illness in later life, damaging physical, psychological, and social functions [5,16,17]. Most studies have shown that depression tends to increase with age [18] and is higher among those who live with their families [19,20] and those who are less educated [21]. A meta-analysis has also indicated risk factors for depressive symptoms among older Chinese adults [22]. However, these studies have focused on the relationship between physical symptoms and depression level or factors associated with depressive symptoms. There is a paucity of research on depression level and predictive factors among CDEP in Taiwan. Therefore, this study sought to explore the predictors of depression level in Taiwan’s CDEP so as to (1) better understand the current situation of depression prevalence and severity among the elderly in the community and (2) explore predictors that affect the depression level among CDEP.

## 2. Methods

### 2.1. Research Design

This cross-sectional study used a convenience sampling method and structured questionnaires collected from January 2019 to December 2019 from three communities in North Kaohsiung of Taiwan. 

### 2.2. Participants

The eligibility criteria for inclusion were: (1) age over 65 years and living in a community; (2) free of mental illness, dementia, or other psychological conditions; (3) ability to communicate with the researcher in Mandarin, Taiwanese, or Hakka languages; and (4) voluntary agreement to participate in the study after its purpose was explained. 

According to Harris [23], for regression equations using six or more predictors, an absolute minimum of 10 participants per predictor variable is appropriate. With 11 predictors, at least 110 participants would be needed. To account for invalid questionnaires, an additional 30% of samples were collected for this study, making the total number of participants 150.

### 2.3. Instruments

#### 2.3.1. Demographics

The following demographic information was collected: age, gender (male or female), education level (illiterate, elementary school, junior high school, senior high school, or college degree or above), marital status (having spouse or no spouse), religious beliefs (yes or no), living status (living alone or living with family), employment status (yes or no), and the number of chronic diseases.

#### 2.3.2. Daily Life Function

Daily life function was measured using the Functional Independence Measure (FIM) [24]. This scale was designed to measure the degree of disability and assess the level of help required by an individual to perform activities of daily living. The FIM is composed of 13 items that assess motor function based on scores on a seven-point Likert scale (1–7 points), with higher scores indicating better daily life function. The FIM showed good internal consistency in this study, with a Cronbach’s alpha of 0.99.

#### 2.3.3. Interpersonal Support Evaluation List (ISEL)

This scale was used to assess the perceived availability of assistance from spouses, family members, relatives and friends, medical staff, colleagues and others. The original scale by Cohen and Syme [25] contained the four dimensions of substantial help, information support, emotional support, and self-esteem, scored on a four-point Likert scale (0–3 points), with higher scores indicating better social support. Chen et al. [26] translated the 16 items into Chinese. The two-week retest reliability was 0.77, and the internal consistency Cronbach’s α was 0.81. In this study, the Cronbach’s α coefficient was 0.82.

#### 2.3.4. UCLA Loneliness Scale

This study adopted the third edition of the UCLA Sense of Loneliness Scale revised by Russell [27], as translated into Chinese by Chang and Yang [28]. The CVI value was 0.85, and its reliability was tested for the elderly, with a Cronbach’s α value between 0.82 and 0.89. There are 20 questions on this scale, which are scored using a four-point Likert scale, ranging from 1 (never) to 4 (always), resulting in a total score between 20 to 80 points, with higher scores indicating greater loneliness. In the present study, Cronbach’s α was 0.88.

#### 2.3.5. Geriatric Depression Scale Short Form (GDS-SF)

The Geriatric Depression Scale (GDS) was developed by Brink et al. in the United States, from which a short form (GDS-SF) was developed [29]. The original questionnaire’s internal consistency Cronbach’s α coefficient was 0.72. It is widely used for psychological evaluation to screen the depression level of the elderly in long-term care institutions, without racial or regional bias, and has high reliability and validity in research around the world. The questionnaire has 15 dichotomous “yes” (1 point) and “no” (0 points) items related to emotional, cognitive, and behavioral symptoms to evaluate life over the past week. The total scores are 0–15, with higher scores indicating a higher depression level. The scores are also categorized into normal, mild, moderate, and severe depression levels based on scores between 0–4, 5–8, 9–11, and 12–15, respectively. This study used the translation by Chan [30], which has a Cronbach’s α coefficient of 0.89 for internal consistency and a retest reliability of 0.85. In this study, Cronbach’s α was 0.86.

### 2.4. Study Process

The study was reviewed and approved (IRB No.: EMRP-107-067) by the Institutional Review Board of a medical institution. This study was conducted using convenience sampling, and we collected data from three communities in North Kaohsiung during community screening. The researchers first discussed the purpose, content, and duration with the community leaders and obtained their consent. Researchers went to the village activity locations to collect data. Those who met the eligibility criteria were interviewed individually using a one-on-one questionnaire. The interview time allotted for each participant was approximately 15–25 min. After the interview, a small gift was given to the participant.

### 2.5. Data Analysis

After confirming that there were no omissions by going through the questionnaire content item by item, the data were coded and analyzed using SPSS 22.0. The collected data were analyzed using descriptive statistics including frequency distribution, percentage, mean, and standard deviation, while inferential statistics were analyzed using multiple regression analysis. Statistical significance was set at *p* < 0.05.

## 3. Results

### 3.1. Demographic Characteristics of the Community-Dwelling Elderly Persons

The subjects were mainly in the age group of 65–74 years, with an average age of 73.4 years; women accounted for 61.3%; primary school graduates accounted for 41.3%, followed by illiterates at 21.3%. In terms of marital status, 85.3% of the participants had a spouse. The presence of religious beliefs accounted for 92.7%, living with family accounted for 91.3%, and no employment accounted for 78.7%. The average number of chronic diseases was 1.12, the average score of daily life was 87.93, and the average total score of social support was 34.57. The average total score for loneliness was 38.79, indicating moderate loneliness (Table 1).

### 3.2. Depression Level of Community-Dwelling Elderly Persons

The average total score for depression among the elderly was 3.12. Participants with different levels of depression accounted for 26%. The percentages of participants with mild, moderate, and severe depression were 12.0%, 7.3%, and 6.7%, respectively (Table 2).

### 3.3. Predictors of Depression Level among the Community-Dwelling Elderly Persons

Table 3 shows that after adjusting for the demographic variables, daily life function, social support, and loneliness, the factors affecting the depression level among the CDEP were education level, living status, chronic disease, daily life function, social support, and loneliness, which could significantly predict the variation of depression by 63.4%. This means that the depression level among the CDEP correlated with the education level (junior high school vs. illiterate, β = −1.67, 95%, CI = from −3.21 to −0.13, *p* = 0.035; college or above degree vs. illiterate, β = −2.85, 95% CI = from −5.29 to −0.42, *p* = 0.023), living with family members (β = 2.28, 95% CI = 0.74–3.83, *p* = 0.004), chronic disease (β = 0.66, 95% CI = 0.26–1.07, *p* = 0.002), worse daily life function (β = −0.05, 95% CI = from −0.09 to −0.02, *p* = 0.002), social support (β = −0.15, 95% CI = from −0.23 to −0.08, *p* < 0.001), and loneliness (β = 0.11, 95% CI = 0.03–0.18, *p* = 0.006) (Table 3).

## 4. Discussion

This study explored the depression level and related factors of CDEP. The elderly female population in the sampled community was larger than that of the males, and most were living with their families. These results are similar to those presented by the Civil Affairs Bureau of the Kaohsiung City Government in 2021. The average depression level in this elderly population was 3.12, and the prevalence of depression was 26%. Among the CDEP with depression, 12.0% had mild depression, and 6.7% had severe depression. This finding is similar to that of a multicenter survey in China [21]. 

This study found that the education level of the CDEP was negatively correlated with the depression level, consistent with Conner et al. [31] and Zhong et al. [21]. Individuals with higher education are more willing to participate in learning activities, continuing education, and engaging in volunteer service and leisure activities. Through these activities and volunteering, the elderly persons regain their self-esteem, self-confidence, and self-efficacy, as well as develop a sense of belonging, leading to a sense of being needed and contributing to the community. In addition, these activities can support the physical and mental health of the elderly to slow down the aging process, reduce mortality, and alleviate loneliness and depression [32,33]. 

It was found that a greater depression level among the CDEP correlated with living with family members. This result is consistent with those of other studies [18,34]. The elderly who live with their families may still feel lonely because their families cannot meet their needs, which increases their isolation and depression [19,20]. It also suggests that poor family functioning can hinder family relationships and affect the physical, emotional, and mental health of the elderly, resulting in symptoms of depression. 

This study found that CDEP with more chronic disease were more likely to be depressed, consistent with the findings of Liu et al. [35] and Tragantzopoulou and Giannouli [36]. CDEP with physical and psychological disorders that affected their daily life had higher levels of depression. Our study also found that engagement in activities of daily living was negatively correlated with the depression level. This is consistent with the results of Kim et al. [15] and Chen [26] that many elderlies can care for themselves independently in daily life, participate in voluntary services or social activities, and receive support from relatives and friends, so they will be physically and mentally more active and thus have higher life satisfaction and lesser loneliness and depression. 

Social support for the CDEP was significantly negatively correlated with depression in this study. The more inadequate the social support function, the greater the depression. Good overall social support, living with a spouse or a partner, having an appropriate social network, having more connections with family and friends, good support from family members, and satisfaction with social support have all been found to be correlated with lower depression symptoms [37]. The elderly who promote social participation, maintain social connections, and establish new social networks (with children, relatives, and friends) have less depression [38]. This indicates that CDEP engagement in emotional social activities, including chatting with acquaintances, visiting family or close friends, and participating in religious and leisure activities, leads to lesser depression. 

This result shows a positive correlation between loneliness and depression among the elderly in the community. Social isolation and depression may have profound negative effects on mental health, including anxiety, stress, and insomnia; loneliness, in terms of a sense of worthlessness, is significantly positively correlated with depression [12]. According to Donovan and Blazer [39], social disengagement puts individuals at greater risk of disease morbidity and mortality at a rate that is higher than or equal to that of traditional risk factors such as alcohol consumption and obesity, and this has profound effects on mental health. The association between loneliness and depression indicates that loneliness can exacerbate depression.

## 5. Limitations and Recommendations

This study had several limitations. First, convenience sampling may have affected the representativeness and generalizability of the results. Second, this was a cross-sectional study in which a causal relationship could not be identified. Third, there is a possibility of recall bias due to the use of self-reported measures. Based on these limitations, we have some recommendations for further research. First, random sampling should be used to avoid selection bias and to improve the representativeness and generalizability of the findings. Second, depression status and related factors of the elderly in the community can be tracked over a longer period to understand the causal relationship. Third, the objective measures can be increased to evaluate the depression level and confirm the findings. Fourth, the study was conducted before the outbreak of COVID-19, therefore, participants have not been influenced by the panic. Future studies can further discuss the influences of COVID-19 on the depression of CDEP, including the health behaviors, psychological, social, and environmental factors of depression.

## 6. Conclusions

This study found that participants with different levels of depression accounted for 26%, and education level, living status, chronic disease, daily life function, social support, and loneliness predicted 63.4% of the variation in the depression level among the CDEP. The current study adds to the emerging literature highlighting that loneliness can exacerbate depression, whereas social support negatively affects depression. However, our study also found that elderly people who live with their families may still feel lonely because their families may not meet their needs. Therefore, the role of social support becomes more important, and this indicates that if CDEP can engage in social activities and receive support from relatives and friends, they will experience less depression. Thus, clinical staff should understand empirical data pertaining to the factors related to depression that impact the physical and mental health of the CDEP and proactively provide consultations to encourage the elderly to participate in volunteering activities, find like-minded friends, expand their social networks, and participate in leisure activities. This can reduce feelings of loneliness and depression, thereby enhancing the sense of value and meaning in life. It is also suggested that future studies should include interventional research on “elderly self-health management” and “interpersonal interaction and social activities” for remote home care to understand its effect on remote care of depression among the CDEP.

## Figures and Tables

**Table 1 ijerph-19-09414-t001:** Demographic characteristics of the community-dwelling elderly persons (*N* = 150).

Variable	*N* (%)	*M* (*SD*)
**Demographics**		
**Age**		73.4 (6.9)
**Gender**		
male	58 (38.7)	
Female	92 (61.3)	
**Educational level**		
Illiterate	32 (21.3)	
Elementary school	62 (41.3)	
Junior high school	20 (13.3)	
Senior high school	30 (20.0)	
College or above degree	6 (4.0)	
**Marital status**		
Has spouse	128 (85.3)	
No spouse	22 (14.7)	
**Religious belief**		
Yes	139 (92.7)	
No	11 (7.3)	
**Living status**		
Living alone	13 (8.7)	
Living with family	137 (91.3)	
**Employment status**		
No	118 (78.7)	
Yes	32 (21.3)	
**Chronic disease**		1.12 (1.16)
**Daily life function**		87.93 (13.18)
**Social support**		34.57 (9.20)
**Loneliness**		38.79 (9.62)

SD = standard deviation.

**Table 2 ijerph-19-09414-t002:** Depression level of community-dwelling elderly persons (*N* = 150).

Variable	*N* (%)	*M* (*SD*)
Depression level		3.12 (3.97)
No (0–4 points)	111 (74.0)	
Yes (5–15 points)	39 (26.0)	
Mild (5–8)	18 (12.0)	
Moderate (9–11)	11 (7.3)	
Severe (12–15)	10 (6.7)	

SD = standard deviation.

**Table 3 ijerph-19-09414-t003:** Predictors of depression level in community-dwelling elderly persons (*N* = 150).

Independent Variable	Crude β (95% CI)	*p* Value	Adjusted β (95% CI)	*p* Value
**Demographics**				
Age	0.16 (0.07–0.25)	0.001	−0.01 (−0.08–0.06)	0.806
Gender				
Male	Ref.		Ref.	
Female	−0.03 (−1.34–1.28)	0.965	−0.49 (−1.41–0.44)	0.306
Education level				
Illiterate	Ref.		Ref.	
Elementary school	−1.97 (−3.63–−0.31)	0.021	−0.95 (−2.12–0.23)	0.116
Junior high school	−2.81 (−4.98–−0.64)	0.012	−1.67 (−3.21–−0.13)	0.035
Senior high school	−2.14 (−4.07–−0.21)	0.032	−0.83 (−2.26–0.60)	0.258
College or above degree	−4.24 (−7.63–−0.85)	0.015	−2.85 (−5.29–−0.42)	0.023
Marital status				
Has spouse	Ref.		Ref.	
No spouse	1.08 (−0.71–2.88)	0.238	0.87 (−0.34–2.08)	0.163
Religious belief				
Yes	Ref.		Ref.	
No	2.03 (−0.40–4.45)	0.103	1.34 (−0.36–3.04)	0.124
Living status				
Living alone	Ref.		Ref.	
Living with family	1.23 (−1.03–3.49)	0.289	2.28 (0.74–3.83)	0.004
Employment status				
No	Ref.		Ref.	
Yes	−1.70 (−3.23–−0.17)	0.031	−0.88 (−1.93–0.17)	0.102
Chronic disease	1.33 (0.83–1.84)	<0.001	0.66 (0.26–1.07)	0.002
**Daily life function**	−0.11 (−0.16–−0.07)	<0.001	−0.05 (−0.09–−0.02)	0.002
**Social support**	−0.29 (−0.35–−0.24)	<0.001	−0.15 (−0.23–−0.08)	<0.001
**Loneliness**	0.27 (0.22–0.32)	<0.001	0.11 (0.03–0.18)	0.006

## Data Availability

The data presented in this study are available on request from the first author, C.-C.P. The data are not publicly available due to privacy.

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
