# Peer review of "Predictors of Depression Level among Community-Dwelling Elderly Persons"

_ijerph, 2022, doi:10.3390/ijerph19159414_

Round 1

Reviewer 1 Report

I believe the manuscript has been significantly improved and
now warrants publication in ijerph.

Author Response

Authors’ response:

Thank you very much.

Reviewer 2 Report

This is a small community study that addresses an important public health issue.  Although it seems that most parts of the study were done properly, it has some minor problems, and a major problem in the analyses/results and the related interpretations. Below are some comments:

Depression in community dwelling older adults is not a new topic.  Can the authors elaborate how this study adds to the literature?

The study examined depressive symptoms rather than clinical depression as an outcome. Please revise the whole paper as appropriate.

 2.1 Research design (p.2) and 2.4 Study process (p. 4): To what extent was the 3 communities sampled representative of the population in Taiwan? Please elaborate and provide data as feasible.  Also, please provide details on the community screening in which participants were screened and recruited.  By “research conditions”, did the authors mean eligibility criteria? Please provide details.

2.2 Participants (p.2): In the power analyses the authors assumed an effect size of 0.20. Which variable was used to come up with this estimated effect size, and what is the reference?

2.5 Data analysis and 3. Results (p.4):  Did the authors look at the additional variance accounted for by loneliness as the key predictor only?  It seems that 63.4 % was the variance accounted for all predictors in the model.  If that is indeed the case, please revise Table 3 (and the corresponding texts) to show the changes in the regression results when the predictors are added in a stepwise fashion. It’d be necessary to look at the variance accounted for by the various predictors after adjusting for the demographic variables before the authors can draw conclusions about the results.

Author Response

Reviewer2#

This is a small community study that addresses an important public health issue. Although it seems that most parts of the study were done properly, it has some minor problems, and a major problem in the analyses/results and the related interpretations. Below are some comments:

Depression in community dwelling older adults is not a new topic. Can the authors elaborate how this study adds to the literature?

Authors’ response:

Thank you very much. This study adds to the emerging literature highlighting loneliness can exacerbate depression, whereas social support negatively affects depression. However, our study also finds that elderly people who live with their families may still feel lonely because their families may not meet their needs. Therefore, the role of social support becomes more important, and this indicates that if CDEP can engage in social activities and receive support from relatives and friends, they will experience less depression. We have elaborated this and provided details on the Conclusions section of this revised manuscript.

The study examined depressive symptoms rather than clinical depression as an outcome. Please revise the whole paper as appropriate.

Authors’ response:

Thank you very much for the suggestions. We have rechecked the whole paper as appropriate, revised the title to fit the content, highlighted in blue and used the Track Change.

2.1 Research design (p.2) and 2.4 Study process (p. 4): To what extent was the 3 communities sampled representative of the population in Taiwan? Please elaborate and provide data as feasible. Also, please provide details on the community screening in which participants were screened and recruited.  By “research conditions”, did the authors mean eligibility criteria? Please provide details.

Authors’ response:

Thank you very much for the comments. Since the 3 communities were screened and recruited from North Kaohsiung of Taiwan, one limitation of the current work was the use of convenience sampling, which may limit the generalizability of the findings. We have pointed out this limitation in “Limitations and recommendations”.

We have elaborated and provided data as feasible. In addition, we provided details on the community screening in which participants were screened and recruited in “Study process”. We also revised the “research conditions” to “eligibility criteria”.

2.2 Participants (p.2): In the power analyses the authors assumed an effect size of 0.20. Which variable was used to come up with this estimated effect size, and what is the reference?

Authors’ response:

Thank you for your comments. We revised the 2.2 participants section of this manuscript according to the criteria supplied by Harris, R. J. (1985). A primer of multivariate statistics (2nd ed.). New York: Academic Press.

2.5 Data analysis and 3. Results (p.4):  Did the authors look at the additional variance accounted for by loneliness as the key predictor only?  It seems that 63.4 % was the variance accounted for all predictors in the model.  If that is indeed the case, please revise Table 3 (and the corresponding texts) to show the changes in the regression results when the predictors are added in a stepwise fashion. It’d be necessary to look at the variance accounted for by the various predictors after adjusting for the demographic variables before the authors can draw conclusions about the results.

Authors’ response:

Thank you very much for the comments. We have revised the title to fit the contents, and heighted in blue using Track Change.

Round 2

Reviewer 2 Report

The authors have addressed my previous comments satisfactorily. This revision has improved significantly. 

This manuscript is a resubmission of an earlier submission. The following is a list of the peer review reports and author responses from that submission.

Round 1

Reviewer 1 Report

I would like to thank you for the opportunity to review this manuscript.

The authors examined the association of demographics, daily life, social support, and loneliness with depression levels among community-dwelling older adults in Taiwan. The topic is important and interesting. However, there are a series of major issues that impact the overall quality of the manuscript.

  1. I would suggest the authors rework on the introduction section in terms of the following aspects:

(1) the overall structure should be revised, e.g., first introducing the research focus and stating the problem, then reviewing relevant literature and indicating the research gap, and finally describing the aim of the current study and its significance.

(2) The authors said “there is a lack of research on depression in Taiwan’s community-dwelling older adults”, but I just quickly screened the literature and found there have been a lot of studies examining the depression in older adults in Taiwan (e.g., Lai et al., 2020; Qiu et al., 2020; Huang et al., 2021). In such a case, I think a more comprehensive literature review is needed.

*Lai, H. C., Hsu, N. W., Chou, P., & Chen, H. C. (2020). The associations between various sleep-wake disturbances and depression in community-dwelling older adults-the Yilan study, Taiwan. Aging & Mental Health24(5), 717-724.

*Qiu, Q. W., Qian, S., Li, J. Y., Jia, R. X., Wang, Y. Q., & Xu, Y. (2020). Risk factors for depressive symptoms among older Chinese adults: A meta-analysis. Journal of Affective Disorders277, 341-346.

*Huang, M. H., Wang, Y. P., Wu, P. S., Chan, Y. L. E., Cheng, C. M., Yang, C. H., ... & Tsai, C. F. (2021). Association between gastrointestinal symptoms and depression among older adults in Taiwan: a cross-sectional study. Journal of the Chinese Medical Association84(3), 331-335.

(3) I would also suggest the authors improve the overall logic of the introduction. A potential way is to divide the study variables into different categories, e.g., demographic, psychological, and social correlates of depression.

(4) The significance and importance of this study need to be explicitly indicated.

(5) I would suggest the authors reconsider the study aims and hypotheses. The second objective and third objective are overlapped, as both of them are examining the association between variables.

  1. More elaboration is needed for the methods part:

(1) changing “a cross-sectional correlation study” to “a cross-sectional study”;

(2) supplementing a citation to support the sample size estimate.

(3) supplementing a couple of sentences to describe the procedure of the survey study, e.g., how did the authors disseminate the recruitment information, how did the survey deliver (e.g., face to face or online), what was the duration for each survey? Anyway, I would suggest the authors strictly follow the STROBE checklist and add the study process diagram.

(4) moving the data analysis to the end of the methods part. Also, the Pearson correlation is inappropriate for binary and categorical variables’ correlation tests. Statistical significance level should be indicated as well.

(5) For demographic variables, the response options should be added to each variable.

(6) For the GDS-SF, scores of 0-4 are considered normal, depending on age, education, and complaints; 5-8 indicate mild depression; 9-11 indicate moderate depression; and 12-15 indicate severe depression. I am wondering what is the percentage of participants indicating significant depressive symptoms (i.e., scoring >=5) in the study samples, as this would be of interests for many readers.

In this paper, the authors used the mean score of the depression. I think it is fine to report the values of depression, but in such a case, it is not proper to say the outcome variable is depressive symptoms. As the term of “depressive symptoms” is commonly used as a categorical concept, i.e., yes/no, or no/mild/moderate/severe depressive symptoms. Given this, I would suggest the authors using the term of “depression levels” instead of the descriptors “depressive symptoms” throughout the manuscript.

  1. For the results part, age also showed significant correlation with depression levels, I am wondering why the age was not included in the regression model.

Further, following my previous comment, I would suggest the authors conduct multi-level regression to examine the association of different blocks of factors with older adults' depression levels, e.g., adding demographic block in the first model, then adding the psychological factors block to the second model, and finally adding the social factors block to the third model.

In addition, I am wondering if there were interaction effects between these independent variables? Did the authors test the moderating role of some factors in the study associations? (e.g., Liang et al., 2021a)

*Liang, W., Duan, Y., Shang, B., Hu, C., Baker, J. S., Lin, Z., ... & Wang, Y. (2021a). Precautionary behavior and depression in older adults during the COVID-19 pandemic: an online cross-sectional study in Hubei, China. International journal of environmental research and public health18(4), 1853.

  1. For the discussion part, the influence of the COVID-19 pandemic could be discussed (e.g., Liang et al., 2021a, Liang et al., 2021b, Yan et al., 2022; Huang & Zhao, 2020). Also, more variables, such as health behaviors, more psychological correlates, health services, relevant policy could be considered in the future studies.

*Liang, W., Duan, Y., Yang, M., Shang, B., Hu, C., Wang, Y., & Baker, J. S. (2021b). Behavioral and Mental Responses towards the COVID-19 Pandemic among Chinese Older Adults: A Cross-Sectional Study. Journal of Risk and Financial Management14(12), 568.

*Yan, Y., Du, X., Lai, L., Ren, Z., & Li, H. (2022). Prevalence of depressive and anxiety symptoms among Chinese older adults during the COVID-19 pandemic: A systematic review and meta-analysis. Journal of geriatric psychiatry and neurology35(2), 182-195.

*Huang, Y., & Zhao, N. (2020). Chinese mental health burden during the COVID-19 pandemic. Asian journal of psychiatry51, 102052.

In addition, the implications of the research findings and more concrete suggestions should be supplmented.

Finally, I would suggest the authors move the limitation parts to the end of the discussion. Also, several limitations should be mentioned, including the representativeness and generalizability issues, recall bias caused by the self-reported measures, also the causal relationship could not be identified by using cross-sectional design.

Hope the above comments help!

Author Response

Dear reviewer,

Thank you very much for the review report and the reviewer's comments for our above-referred manuscript. We have revised the manuscript following your instructions and responded to the reviewer’s comments point-by-point. The copy marked blue and used “Track Changes” of the revised manuscript is attached. We have provided our responses to the reviewer’s comments in the attachment file. We very much appreciate your review of the revised manuscript and consideration for publication in this leading journal.

Sincerely,

Reviewer 2 Report

  • The research problem proposed and discussed by the author is interesting and particularly important for two reasons. First, the growing sense of isolation and depression among the elderly persons. The second important aspect of the study is the relevance of the results to psychological counseling and therapeutic practice dedicated to elderly people.
  • Abstract lacks the information about the sample (e.g., the numbers of men and women, and average and SD of age).
  • The theoretical context could have been a bit more elaborate and based on theory instead of solely relying on empirical findings.
  • Considering the obvious conclusions from the previous studies (see lines 104-108), it is difficult to see your study as an explorative one. Therefore, you should formulate the hypotheses and present them at the end of the Introduction.
  • Given the non-experimental method, the authors should be careful not to claim causality and rephrase some causal wording (eg. lines: 25, 29, 59,199, 233).
  • Convenience sampling was adopted in the study. The authors should describe more precisely how the sample was recruited so that the reader can better judge how this sampling method could have potentially affected the results.
  • In Data analysis section you mentioned about using Pearson correlation. However, please note that it is inappropriate to apply Pearson when using nominal variables (see Table 2 and some correlations you reported there, e.g. Gender, living status, marital status). In this case, you should use Point-biserial correlation or Eta coefficient.
  • Correlation and regression analyzes should be supplemented with 95% confidence intervals
  • Discussion is clearly presented and nicely balanced in addressing present results, past findings, future research, as well as limitations.

Author Response

(The authors gave the same response as above.)

Round 2

Reviewer 1 Report

I am satisfied with some revisions of the paper, whereas I find several key points have not been well addressed.

1. The logic and rationale of doing the current study could be further improved (e.g., the rationale for selecting those influential factors).

2. I do not think the sample size of 150 could reflect the general prevalence of depression issue.

3. The study was conducted before the outbreak of the COVID-19 and individuals have been considerably impacted by the pandemic including the mental status (social event has been evident to be critical correlates of depression in previous evidence). This should be discussed (see my suggestions in review round 1).

4. How about the overall effect size of the prediction of these factors in the authors' model?

Additionally, some key determinants (e.g., health behaviors, psychological, social, and environmental factors) of depression have not been examined in this study, which should be at least mentioned (see my suggestions in review round 1).

5. The limitation part should be further improved e.g., some repeated points.